# Spatial and Multicriteria Analysis of Dimension Stones and Crushed Rocks Quarrying in the Context of Sustainable Regional Development: Case Study of Lower Silesia (Poland)

**Jan Blachowski ***[ID] **and Anna Buczyńska**[ID]

Faculty of Geoengineering, Mining and Geology, Wrocław University of Science and Technology, 50-370 Wroclaw, Poland; anna.buczynska@pwr.edu.pl
* Correspondence: jan.blachowski@pwr.edu.pl; Tel.: +48-71-320-68-75

**Abstract:** This study aimed to analyze and assess the spatial and temporal trends in distribution of reserves and production of igneous and metamorphic dimension stones and crushed rocks in the Lower Silesia region, which with 90% of total supply is the key source region of these rock raw materials in Poland. The research utilized descriptive statistics to examine temporal variations of production, as well as to determine sufficiency of these resources for four different scenarios and seven main lithological groups of rocks. Spatial statistics in the geographic information system was used to map changes in spatial distribution of production with the density function, as well as to determine areas of highest concentration in the 2010–2018 period. Then, 169 communes in the region were assessed using the multicriteria analytical hierarchy process (AHP) technique to identify local communities prospective for development of this mining sector. Strong, positive correlation ($r = 0.645$) between year to year production change with annual rate of gross domestic product was determined. Sufficiency of economic reserves was estimated, in four scenarios, as being the lowest for melaphyre and porphyry (25–49 years), and the highest for marble (380–389 years). Reserves of basalt and amphibolite should last for approximately 50 years (32–60 and 36–67 years, respectively), granite for 82 to 110 years, and gabbro and gneiss for over 100 years (78–159 and 76–189 years, respectively). Maps revealed a possible trend of increasing production in quarries located in the central and eastern parts of Lower Silesia, whereas multicriteria analysis allowed for the identification of nine communes with the highest potential for rock raw material quarrying. The practical outcome of this study is a knowledge database for authorities, upon which sustainable management of regional rock raw materials can be based in the context of economic, social, and environmental impacts of their extraction.

**Keywords:** dimension stones; crushed rocks; spatial statistics; spatial analysis; analytical hierarchy process; Poland; Lower Silesia; economy

---

## 1. Introduction

Dimension stones and crushed rocks (DSCR) include igneous rocks such as basalt, diabase, gabbro, granite, melaphyre, porphyry, and syenite; metamorphic rocks such as amphibolite, gneiss, hornfels, metamorphic slate, marble, migmatite, serpentinite, and greenstones; and sedimentary rocks such as chalcedonite, dolomite, quartzite, slate, marl, sandstone, limestone, greywacke, travertine, and conglomerate. According to the United States Geological Survey [1], the main types of dimension (ornamental) rocks include granite, limestone, sandstone and marble, with the first three and dolomite also being sources of crushed rocks [1]. Other sources indicate the use of slate, gabbro, and travertine,

among others, for ornamental purposes, and list the use of basalt, gabbro, and gneiss for crushed rock production [2–4].

Globally, dimension stones and crushed rocks are considered to be key resources, that is, resources that enable proper functioning of the economy and satisfy the living standards of the society. Demand for these rock raw materials, due to their properties being used in the construction industry, including in building, road and railroad industries, construction, and modernization, is related to the growth of the economy. Thus, it is essential to ensure sustainable consumption and production patterns of these natural resources in accordance with the United Nations' Sustainable Development Goals Agenda, including goal 12 of responsible consumption and production (www.un.org/sustainabledevelopment) through responsible territorial development policies. According to the British Geological Survey study, production of DSCR between 2013 and 2017 increased by 8.6% from 1,110,895.7 thousand Mg to 1,206,066.4 thousand Mg in European countries [5]. Poland was the seventh largest producer of dimension stones and crushed rocks in 2017 and the third largest of all aggregates. Despite available resources of domestic dimension stones and crushed rocks, it is important to ensure that economic use of these rock raw materials is carried out in a thoughtful and balanced manner and to secure sufficient supply of these resources that meets the present and provides for future needs [6]. This is due to the fact that the available deposits of DSCR are distributed irregularly and numerous undeveloped deposits are located in areas covered by various forms of nature protection that limit or prohibit their use. In Poland, as a whole, the sufficiency of all types of dimension stones and crushed rock resources is estimated at approximately 50 years with the current level of mining [7]. Research by Lewicka and Burkowicz revealed that, depending on the region, between 41% and 62% of undeveloped deposits are located in protected areas [8]. The Lower Silesia region, because of advantageous geological conditions [8–10], is the principal source and leading supplier of igneous and metamorphic rock raw materials in Poland. The production volume of these DSCR in Lower Silesia compared with the rest of Poland, together with documented geological reserves of these resources, is shown in Figure 1. The size of symbols used depicts total production in 2018. Lower Silesia is the main contributor supplying between 43% and 46% of total DSCR production and approximately 90% of igneous and metamorphic DSCR in the country annually [11]. Therefore, rock raw materials quarrying, and the associated processing and transport sectors have an important part in the national and regional economies and for local communities. In turn, increased demand for these materials leads to environmental and social conflicts of mining and transport.

The two main goals of this study were to analyze and to assess the most recent spatial and temporal changes in distribution of quarrying of igneous and metamorphic dimension stones and crushed rocks in the Lower Silesia region in the 2010–2018 period, as well as to identify areas (communes) with the greatest potential of using the available rock raw material resources for their economic benefit with respect to the existing environmental, social, and infrastructure constraints. For this purpose, the geospatial and descriptive statistics methodology in geographic information systems (GIS) and multicriteria analytic hierarchy process (AHP) technique were adopted. The present study is a consequence of research started by Blachowski and reported in [12], and provides information for regional authorities responsible for spatial development policies. To the best of the authors' knowledge, it is the first work using this multifaceted approach. Table 1 contains a summary review of publications focusing on the topic of rational management of rock raw materials (including dimension stones and crushed rocks), with a list of subjects, methodologies, and resources analyzed in the cited research. The main and most frequent subjects of investigations have focused on environmental issues and include valorization and sustainable use of deposits, social and environmental influence of mining, and management of waste from quarrying of rock raw materials. However, the most common methodological approaches utilize GIS-based mapping and map overlay, as well as a variety of multi criteria decision-making methods.

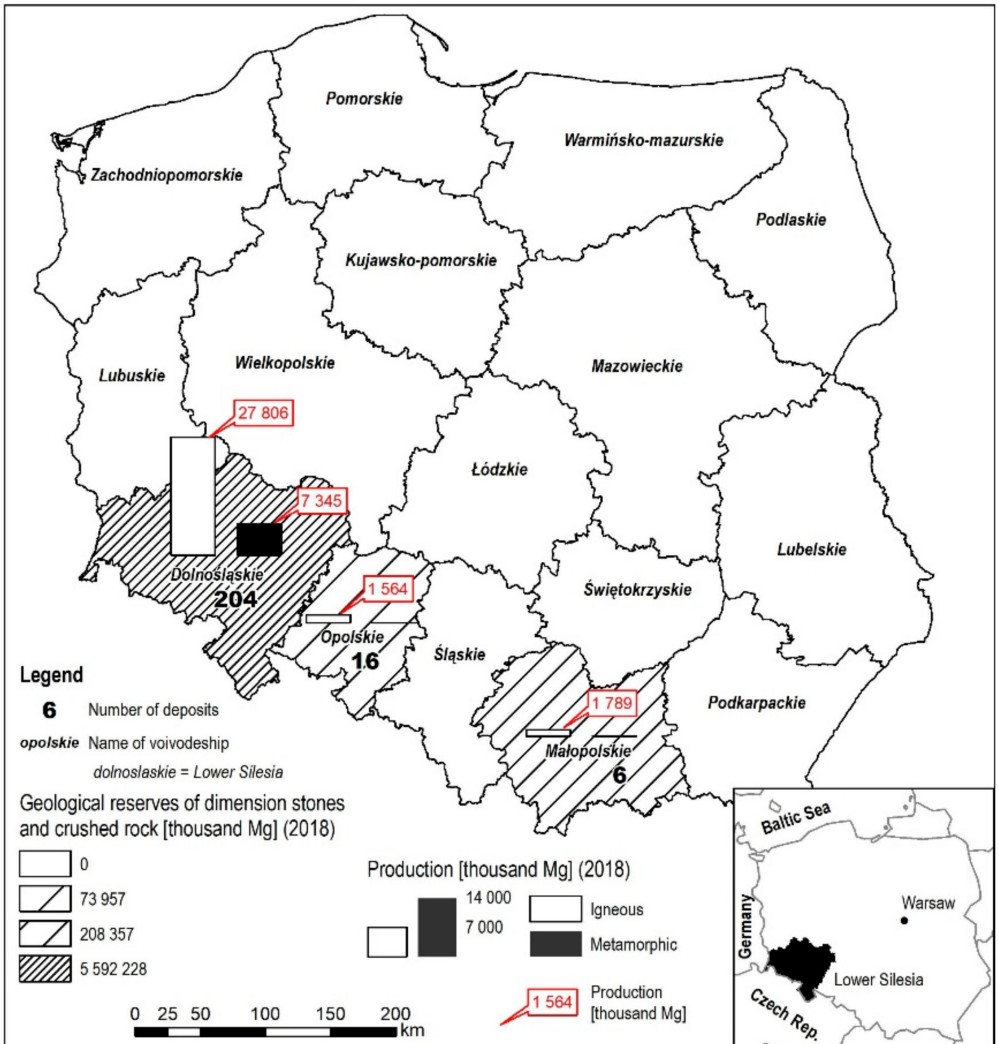

**Figure 1.** Reserves and production of igneous and metamorphic dimension stones and crushed rocks in Lower Silesia and other regions of Poland, based on PGI data [11].

**Table 1.** The topic and methodology applied in reviewed studies related to rock raw materials quarrying and management.

| No. | Reference | Minerals Analyzed (Study Location) | Methodology | Subject of Research | Classification of Research Subject |
|---|---|---|---|---|---|
| 1. | [13] | Metamorphic rocks (Sri Lanka) | GIS-based analysis of cumulative weighed indexes | Identification of most suitable quarry sites | Valorization and sustainable use of deposits of rock raw materials |
| 2. | [14] | Limestone and dolomite (Morocco) | GIS-based weighted criteria overlay analysis | Classification of potential mining sites | |
| 3. | [15] | Aggregates (New Zealand) | GIS and fuzzy logic overlay of resource criteria maps | Aggregate resource opportunity modelling | |
| 4. | [16] | Limestone and dolomite (Greece) | GIS vector map overlay | Suitability of potential crushed rock resource areas for mining | |
| 5. | [17] | Dimension stones and crushed rocks (Poland) | GIS-based multicriteria analysis | Assessment of deposit accessibility for potential use | |
| 6. | [18] | Aggregates (Europe) | LCC (Life Cycle Cost) and LCA (Life Cycle Assessment) | Best available concept (BAC) model for aggregate production and use | Sustainable management of rock raw mineral resources |
| 7. | [19] | Crushed rocks (Poland) | Spatial statistics and interpolation in GIS | Analysis of concentration of crushed rocks quarrying | |
| 8. | [20] | Sand and gravel (USA) | Descriptive statistics, Monte Carlo simulation | Estimation of regional mineral resources | |
| 9. | [12] | Crushed rocks (Poland) | Spatial statistics in GIS | Analysis of spatial distribution of crushed rocks quarrying and transport | Transport of rock raw materials |
| 10. | [21] | Crushed rocks (World) | Literature review | n/a | Review of world literature on the rock raw material quarrying |
| 11. | [22] | Dimension stones (Iran) | AHP-TOPSIS MADM methods and fuzzy logic | Safety ranking of quarries | Safety issues in rock raw materials quarrying |
| 12. | [23] | Sand and gravel (Turkey) | Unsupervised classification of remote sensing data in GIS | Land use change due to open pit mining | Social and environmental impact associated with the extraction of rock raw materials |
| 13. | [24] | Limestone (Italy) | GIS-based spatial interpolation with ordinary kriging technique | Analysis and mapping of acoustic spatial variability generated by quarrying | |
| 14. | [25] | Limestone (Albania) | FOLCHI method | Quantification of the environmental impact of mining | |
| 15 | [26] | Sand and gravel (Oman) | GIS and remote sensing-based rapid impact assessment matrix (RIAM) | Spatial decision support to monitor and evaluate environmental impacts of mining | |
| 16. | [27] | Crushed rocks (Italy) | Descriptive statistics | Strategies for increase of recycled aggregates use | Management of waste from the production of rock raw materials |
| 17. | [28] | Marble (Greece) | Multi-criteria decision analysis (MCDA) and GIS | Evaluation of alternative sites for sustainable disposal of mining waste | |
| 18. | [29] | Rock raw resources (Poland) | Multicriteria analysis with AHP | Assessment of economic value of active mining waste sites | |

## 2. Geology and Mineral Resources of Lower Silesia

The geological structure of Lower Silesia is heterogeneous and varied. This is the result of polyphasic geological evolution that lasted from the upper Proterozoic up to the Quaternary. The complicated geological nature is due to the compilation of processes such as volcanism, block deformations on the foreland of the Alpine orogen, cyclic glaciations, and sea transgressions. The main geologic-tectonic structures run from the NW to SE and are Fore-Sudetic Monocline (in the North of the region), Fore-Sudetic Block, and the Sudetes Mountains in the South, which are separated from the Block with Sudetic Marginal Fault. Some studies also suggest that the latter two are one structure, the Sudetic Block [30]. Extent of the three main geological units has been shown in Figure 2, together with areas of nature protection, where the abbreviation SAC stands for special area of conservation and SPA for special protection areas designated, respectively, under the Habitats Directive and Birds Directive, making up the network of nature protection areas in the European Union.

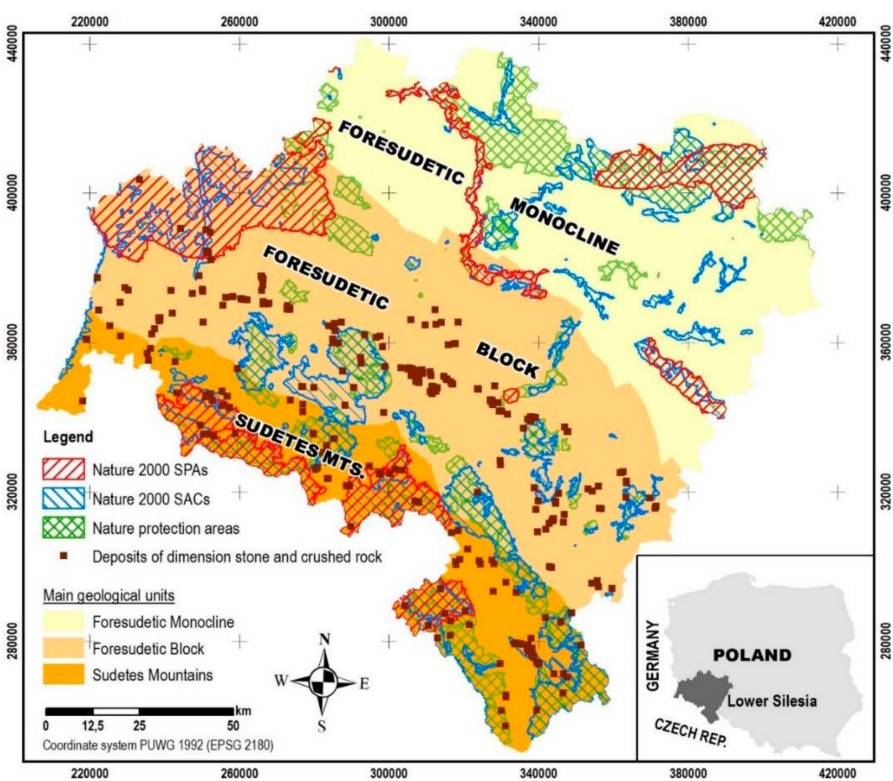

**Figure 2.** Location of dimension stones and crushed rock deposits, main geological units, and nature protection areas in Lower Silesia. SPA denotes special protection area, SAC denotes special areas of conservation.

The Sudetes, in terms of geology, are divided into the Eastern and the Western Sudetes. The structure is composed of various igneous, metamorphic, and sedimentary rocks dating back to the Precambrian to the Cenozoic. These rocks build numerous smaller tectonic units, separated by faults, and together form a heterogenous geological surface.

The Fore-Sudetic Block consists of two structural levels. The older one is built of metamorphic and igneous rocks dating back to the Devonian–Permian period. It is partly covered by deposits of sedimentary rocks from the Permian to Pleistocene making up the younger structural level [31]. The unit is also characterized by heterogenous composition and numerous, secondary rock types have been identified there, including gabbro, granite, and serpentinite massifs; metamorphic; and other structures [32].

The Fore-Sudetic Monocline is composed of thick layers of Permian–Mesozoic origin lying with an unconformity on a folded Paleozoic subsurface. The Permian–Mesozoic deposits generally lie at an angle of several degrees towards the north and northeast.

This geological context is the reason for the rich and diversified mineral resources documented in Lower Silesia. These include metallic minerals (e.g., copper and silver ore), energy minerals (e.g., hard coal and lignite), and various rock minerals [10]. The region is the primary zone of dimension stones and crushed rock of both igneous and metamorphic origin with 96% of national geological and economic reserves [11]. The Lower Silesia has most of the national igneous rocks, such as basalt, granite, melaphyre, and porphyry, as well as the only resources of gabbro and syenite in the country. The same can be said about metamorphic rocks, such as the most of the amphiboles, serpentine, hornfels, migmatite, and marble [33]. The most well-known DSCR from Lower Silesia include the Strzelin and Strzegom granites, as well as white and green Marianna marbles [34]. The basic parameter that distinguishes between dimension and crushed rocks is blockiness, that is, the divisibility of rocks, which enables economically justified operation of blocks [35]. Other parameters include, for instance, durability and the ability of the stone to hold a surface finish [36].

Out of 231 deposits of igneous and metamorphic stones and crushed rocks in Poland, 204 have been documented in the Lower Silesia [11]. These are located in the boundaries of Sudetes and Fore-Sudetic Block; the Fore-Sudetic Monocline has no part in economic reserves of these rock raw materials. The sedimentary DSCR, whose resources and production constitute 1.75% of all DSCR in Lower Silesia, have not been focused upon in this study. Detailed statistics on the reserves of dimension stones and crushed rocks are described in the Results section of this paper.

## 3. Materials and Methods

The study utilized analysis of quantitative and spatial data with descriptive and spatial statistics methods in GIS. For the analysis of the potential of communes for the sustainable development of the rock raw materials industry, the analytical hierarchy process (AHP) methodology and GIS were adopted. The analyzed data include: (i) balance sheets or mineral reserves and mineral production published annually by the Polish Geological Institute [11,37–44], (ii) geospatial datasets with location and extent of mineral deposits from the Polish Geological Institute database, (iii) geospatial datasets with boundaries of administrative units from the National Office of Geodesy and Cartography database, (iv) geospatial datasets with location of transport infrastructure and protected areas of environmental value taken from the Institute for Territorial Development (Lower Silesia Marshal Office), (v) balance sheets of taxes paid by mining companies in accordance with the promulgation of the Minister of Environment regarding mining fee rates [45–51], and (vi) World Bank data on annual rate of gross domestic product (GDP) [52].

### 3.1. Analysis of Dimension Stones and Crushed Rocks Production

Descriptive statistics were used to provide summaries about the production of igneous and metamorphic DSCR raw materials in the Lower Silesia region for the 2010–2018 period. The quantitative summaries include tables and summary statistics, Pearson correlation coefficient test statistics, as well as graphs.

### 3.2. Analysis of Spatial Distribution of Dimension Stones and Crushed Rock Production

The methodology involved calculations of the production density, mean center, and standard deviational ellipse (directional deviation). In the first, density of production was calculated to determine spatial distribution and concentration of quarrying intensity in the region. The density of DSCR

production was calculated with quadratic Kernel weighting function proposed by [53]. The general formula of the function is given by Equation (1):

$$f_\lambda(x) = \frac{1}{n\lambda}\sum K_0\left(\frac{x - x_i}{\lambda}\right),$$

(1)

where

$Ko$ is the Kernel function;

$\lambda$ is the bandwidth (smoothing parameter), which determines the width of search neighborhood.

Quadratic form of the Kernel function is given by Equation (2):

$$K_0(t) = \left\{ \begin{array}{c} 0.75\left(1 - t^2\right) \text{ for } |t| \leq 1 \\ 0 \text{ otherwise} \end{array} \right\},$$

(2)

The input parameters include spatial location ($x$ and $y$ coordinates), population field value, and the search radius [54]. The locations are quarries and the population field are their output (Mg). The search radius for the size of Lower Silesia was set, as in the previous study [8], to 20,000 m. The function calculates surface that represents volume of production per unit area (Mg/km sq). The pixel size of output raster was set to 100 x 100 m [12]. Production density maps for 2010 and 2018 were used to calculate production density difference with raster calculator function.

Then, primary spatial statistics measures were calculated, such as the mean center and standard distance ellipse to quantify concentration of dimension stones and crushed rocks quarrying. The mean center defined the mean $x$ and $y$ coordinates of locations in the study area, whereas standard deviational ellipse measured the standard deviation of the features from the mean center separately for $x$ and $y$ coordinates.

The mean center was calculated for two cases. The first case concerned the unweighted location of all quarries, and the second one weighted location of all quarries, where the weight was defined as annual production. The following formula was used:

$$\bar{x} = \frac{\sum_i (w_i x_i)}{\sum_i w_i}, \bar{y} = \frac{\sum_i (w_i y_i)}{\sum_i w_i}$$

(3)

where

$\bar{x}$, $\bar{y}$ are coordinates of the mean center;

$x_i$, $y_i$ are coordinates of a given location i in the dataset;

$wi$ is the weight of a given location (annual production).

The standard deviational ellipse was calculated for the same two cases, that is, unweighted location of all quarries, and the second one weighted location of all quarries, where the weight was defined as annual production. The following formula was used [55]:

$$SD_x = \sqrt{\frac{\sum_i (x_i - \bar{x})^2}{n}}, SD_y = \sqrt{\frac{\sum_i (y_i - \bar{y})^2}{n}}$$

(4)

where

$SD_x$ is the standard distance for the $x$ axis;

$SD_y$ is the standard distance for the $x$ axis;

$n$ is the number of features (locations).

The weighted standard deviational ellipse was calculated in a similar way, but the squared difference from the mean for each coordinate was multiplied by the weight value before being summed. The greater the distance of the axes, the more the features were dispersed around the center.

### 3.3. Analysis of Dimension Stone and Crushed Rock Available Reserves

Available resources (named in the paper as resource sufficiency) were analyzed in the following way. Total sufficiency for various lithological types of igneous and metamorphic dimension stones and crushed rocks was calculated for four scenarios named A to D. Scenario A was calculated for the production levels for last year's data (2018), scenario B for maximum output, scenario C for minimum output in the analyzed period, and scenario D for an average production (2010–2018). The following formula was used to calculate a particular operation life expectancy in scenario B and to calculate sufficiency of resources for the entire region:

$$S_t = \frac{R_n}{\overline{w_n}} \tag{5}$$

where

$S_t$ is the resource sufficiency in years of the mine, $n$;

$R_n$ denotes economic reserves—technically and economically extractable part of economic resources of the mine, $n$;

$w_n$ is the average annual production of the mine, $n$.

### 3.4. Analysis of the Commune Potential for DSCR Mining Sector

The next part of the study was aimed at assessing the potential of communes with DSCR deposits in the Lower Silesia region to support and develop the rock mining sector of the economy. The selected set of criteria were (i) undeveloped DSCR deposits, (ii) undeveloped DSCR deposits of regional or national significance, (iii) current DSCR quarrying operations, (iv) railroad DSCR loading point infrastructure, (v) revenue from mining taxes, and (vi) potential environmental conflicts (nature protection areas spatially congruent with documented deposits). The potential was assessed with analytical hierarchy process (AHP). The criteria and their weights were determined by a reference group consisting of seven representatives of regional spatial planning (2), science (2), public administration (2), and mining authority (1). The AHP methodology consists of the following steps [56,57]:

(a)　developing the model;
(b)　deriving weights for the criteria;
(c)　checking the consistency;
(d)　deriving local preferences for the alternatives;
(e)　deriving overall priorities and making the final judgement.

Step (a) involves constructing a hierarchy to analyze the decision. The AHP decision model for selecting waste for reuse is structured into a hierarchy of goal, criteria, and alternatives.

Step (b) consists of deriving weights of the criteria, and the importance of criteria is compared pairwise with respect to the desired goal to derive their weights.

In step (c), check of the consistency of judgments is performed; it involves a review of the judgments in order to ensure a reasonable level of consistency in terms of proportionality and transitivity.

Step (d) consists of deriving the local preferences for the alternatives. Priorities for alternatives are derived separately for each criterion following the same method as in step (b). Consistency check is also carried out.

In step (e), overall priorities of the alternatives are calculated, that is, the obtained alternative preferences are combined as a weighted sum to take into account the weight of each criterion. The alternative with the highest overall priority constitutes the best choice.

Thus, if $n$ is the number of analyzed criteria, then the AHP comparison procedure is as follows:

-　Constructing a $n \times n$ pairwise comparison matrix $m$ for analyzed criteria, where *aij* denotes entry in the *i* row and the *j* column of matrix *m*;

-    *aij* states the preference score of criterion *i* to criterion *j* using the nine-integer value scale suggested by [58];
-    Establishing a normalized pairwise comparison matrix *m*, the sum of each column must be equal to 1. This can be obtained using Equation (6) to calculate $\bar{a}_{ij}$ for each entry of matrix $\overline{m}$

$$\bar{a}_{ij} \;=\; \frac{a_{ij}}{\sum_{i\,=\,1}^{n} a_{ij}} \tag{6}$$

-    Determining the relative weights, that is, the average across rows is computed using Equation (7); for each element, the relative weight is within the range of 0 to 1 and a higher weight shows a greater influence of a given element (criterion) [57]:

$$w_i \;=\; \frac{\sum_{i\,=\,1}^{n} a_{ij}}{n} \tag{7}$$

A test of the degree of consistency of the derived weights was performed to check the consistency of the experts' judgements. It involved a calculation of the consistency ratio (CR), which indicates the probability that the matrix values were randomly generated. According to [57], a matrix that has a consistency ratio greater than 0.10 should be re-evaluated. The AHP calculations were performed with the software developed by [59].

## 4. Results

Results of descriptive statistics regarding DSCR resources and quarrying in Lower Silesia were compiled in the form of tables and graphs, whereas results of spatial statistics were presented as maps. The results of multicriteria AHP analysis were presented in tables and on a map to provide the spatial context.

### 4.1. Reserves and Production of Dimension Stones and Crushed Rocks

Geological and economic reserves and production of dimension stones and crushed rock raw materials were analyzed for the 2010–2018 period. These results provide updated and extended information and insight into the changes that have occurred in the region since the analysis described in [12]. Table 2 presents changes of geological and economic reserves in Lower Silesian deposits broken down into the main lithological types of igneous and metamorphic DSCR. In general, available economic reserves in developed deposits increased in Lower Silesia (+2.7%) and, in consequence, also in Poland (+2.5%), as the Lower Silesian reserves constitute 96% of the total. The geological reserves, which also include reserves in currently undeveloped deposits, also increased. However, from analyzing reserves of particular lithological types of igneous and metamorphic DSCR in Lower Silesia, we registered a decrease of economic reserves for three groups of these rock raw materials, whereas reserves of four groups increased. The DSCR groups with increase in economic reserves between 2010 and 2018 were amphibolite (with migmatite and serpentinite) + 59.7%, gneiss +133.7%, marble +122,7%, and granite with syenite +3.1%. The DSCR groups with decrease of economic reserves were basalt −8.2%, gabbro −26.9%, and melaphyre and porphyry −59.6%. The decreases indicated depletion of reserves in operating quarries and potential need to start new mining operations in order to satisfy demand. The decrease of economic reserves was associated with reduction of geological reserves, in cases of basalts and melaphyre with porphyry, which indicated the fact that no new resources of these rock raw materials were documented. In the other groups, new deposits of gneiss and marble contributed to increasing of their geological and economic reserves, the latter of which was associated with new quarry operations. The economic reserves of granite, which had the highest share in total production, remained at a steady level (+3.1%), whereas geological reserves grew by 11.7%. This subject of sufficiency of reserves is further discussed in the following sections.

**Table 2.** Geological and economic reserves of igneous and metamorphic dimension stones and crushed rocks (DSCR) in Lower Silesia and in Poland (2010 to 2018) [11,37–44].

| | Geological Reserves (×1000 Mg) | | Economic Reserves (×1000 Mg) | | Percentage Change (2010–2018) | |
|---|---|---|---|---|---|---|
| | **2010** | **2018** | **2010** | **2018** | **Geological** | **Economic** |
| **POLAND** | **5,679,273** | **5,874,542** | **2,454,688** | **2,515,837** | **3.4%** | **2.5%** |
| **LOWER SILESIA** | **5,369,959** | **5,592,228** | **2,349,981** | **2,413,160** | **4.1%** | **2.7%** |
| Basalt | 572,594 | 535,419 | 366,284 | 336,189 | −6.5% | −8.2% |
| Granite, syenite | 1,820,750 | 2,033,310 | 970,170 | 1,000,061 | 11.7% | 3.1% |
| Gabbro | 512,819 | 527,833 | 367,034 | 268,301 | 2.9% | −26.9% |
| Melaphyre, porphyry | 1,083,505 | 1,054,534 | 332,090 | 134,278 | −2.7% | −59.6% |
| Amphibolite [1] | 286,425 | 296,844 | 67,258 | 107,429 | 3.6% | 59.7% |
| Gneiss [2] | 667,439 | 685,369 | 145,653 | 340,348 | 2.7% | 133.7% |
| Marble | 426,427 | 458,919 | 101,492 | 226,005 | 7.6% | 122.7% |

1—including migmatite, serpentinite, greenstones; 2—including hornfels, metamorphic slate.

In the same period (2010–2018), the total number of documented igneous and metamorphic DSCR deposits (Table 3) decreased by 18. Thus, the decrease of basalt, as well as melaphyre with porphyry reserves, was accompanied by the decrease in number of deposits by −11 and −5, respectively. The number of gabbro deposits remained unchanged, which indicated steady depletion of their reserves. The growth of amphibolite reserves was associated with documentation of new deposits. Table 3, in contrast to the previous study that focused on an earlier period [12], does not include sedimentary rocks.

**Table 3.** Number of igneous and metamorphic DSCR deposits in Lower Silesia (2010–2018) [11,37–44].

| | **2010** | **2011** | **2012** | **2013** | **2014** | **2015** | **2016** | **2017** | **2018** | **Change 2010–2018** |
|---|---|---|---|---|---|---|---|---|---|---|
| **Total** [1] | **222** | **223** | **225** | **224** | **224** | **216** | **203** | **204** | **204** | **−18** |
| Basalt | 47 | 47 | 46 | 44 | 44 | 39 | 36 | 36 | 36 | −11 |
| Granite, syenite | 86 | 86 | 88 | 88 | 88 | 86 | 84 | 85 | 85 | −1 |
| Gabbro | 6 | 6 | 6 | 6 | 6 | 6 | 6 | 6 | 6 | - |
| Melaphyre, porphyry | 26 | 26 | 26 | 26 | 26 | 25 | 21 | 21 | 21 | −5 |
| Amphibolite [2] | 13 | 14 | 15 | 16 | 16 | 16 | 16 | 16 | 16 | +3 |
| Gneis [3] | 22 | 22 | 22 | 22 | 22 | 22 | 20 | 20 | 20 | −2 |
| Marble | 22 | 22 | 22 | 22 | 22 | 22 | 20 | 20 | 20 | −2 |

1—the total number of all DSCR in Lower Silesia in 2018, including sedimentary forms, was 258; 2—including migmatite, serpentinite, greenstones; 3—including hornfels, metamorphic slate.

As it was mentioned earlier, Lower Silesian igneous and metamorphic dimension stones and crushed rocks account for over 90% of the total production in Poland. The actual share for particular lithological groups of these rock raw materials is given in Table 4. The share ranged from 75.9% for melaphyre and porphyry group to 100% for amphibolite, migmatite, and serpentinite groups. On the other hand, production of sedimentary DSCR in Lower Silesia accounted for just 1% to 2% of national supply. This shows that in terms of igneous and metamorphic rock raw materials, the Lower Silesian resources are critical for the national economy, as well as the fact that consumers have to and are willing to pay the costs of transport to locations that are at a greater distance from the source region. This dependence on regional sources is also visible in the time series of annual production (Figures 3 and 4). For the purpose of this study, annual production levels were traced from as early as 1998, and were plotted together with annual change of national gross domestic product (GDP) [52]. The calculated Pearson correlation coefficient that measures the statistical relationship between two continuous variables, that is, annual GDP change and annual change of DSCR production, was equal r = 0.645, which indicates a strong, positive relationship. These results are further discussed later in this paper.

**Table 4.** Production of igneous and metamorphic DSCR in Lower Silesia and in Poland (2018) [11,37–44].

| DSCR Type | Lower Silesia 2018 (×1000 Mg) | Poland 2018 (×1000 Mg) | Lower Silesia Share |
|---|---|---|---|
| Basalt | 7744 | 1307 | 85.6% |
| Granite, syenite | 11,882 | 184 | 98.5% |
| Gabbro | 2859 | 108 | 96.4% |
| Melaphyre, porphyry | 5291 | 1681 | 75.9% |
| Amphibolite [1] | 3016 | 0 | 100.0% |
| Gneiss [2] | 3748 | 70 | 98.2% |
| Marble | 581 | 03 | 99.5% |
| **Igneous and metamorphic DSCR** | **35,121** | **3353** | **91.3%** |

1—including migmatite, serpentinite, greenstones; 2—including hornfels, metamorphic slate.

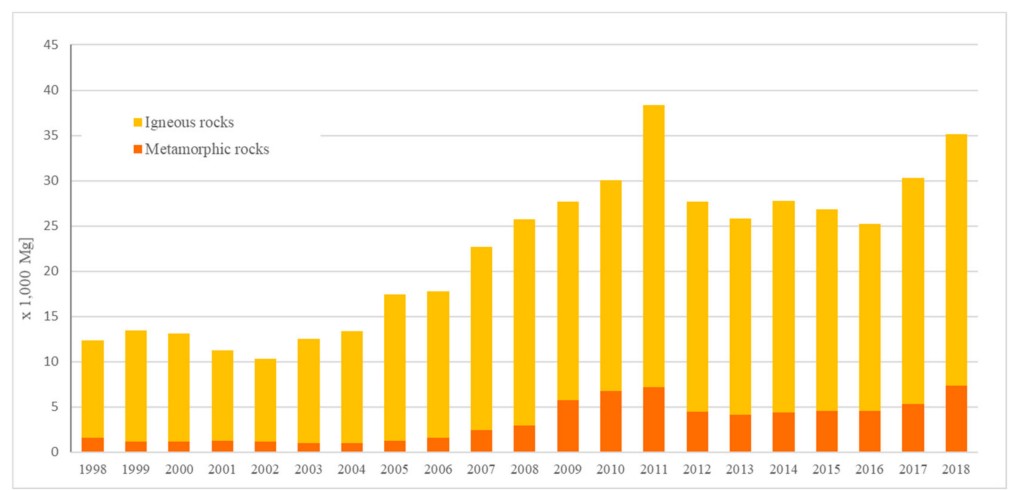

**Figure 3.** Production of igneous and metamorphic DSCR in Lower Silesia between 1998 and 2018 [11,37–44].

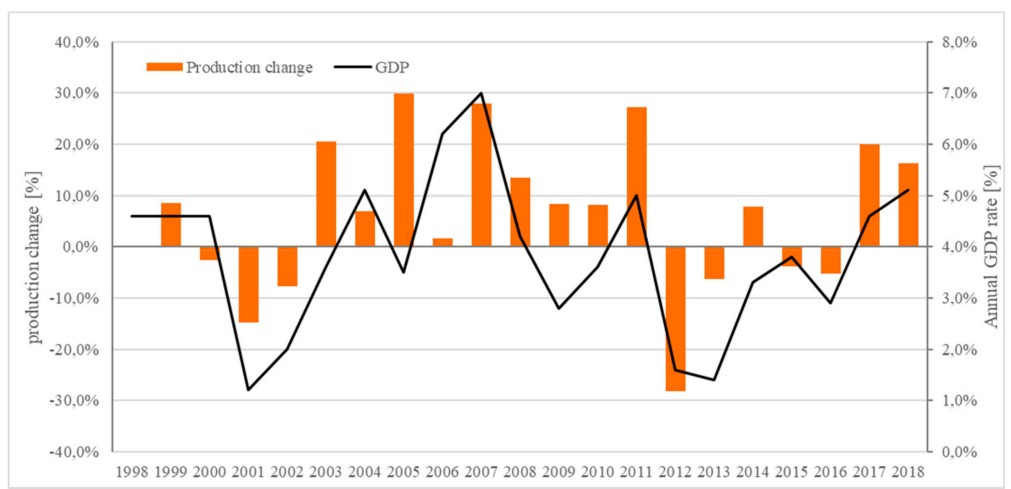

**Figure 4.** Year to year change of igneous and metamorphic DSCR production in Lower Silesia between 1998 and 2018 and annual GDP rate.

### 4.2. Spatial Distribution of Dimension Stone and Crushed Rock Production

GIS spatial statistics were used for analysis of spatial distribution of DSCR production in Lower Silesia and its change in time allowed for identification of areas of the most intensive DSCR quarrying. The map of difference of DSCR production density per unit area between 2010 and 2018 calculated with Kernel density function (Figure 5) indicated areas of intensified DSCR quarrying (red colors)

and areas of quarrying decrease (blue colors). The most intensive quarrying has taken place in the central and south-east parts of the region, whereas peripherally, in relation to Poland, parts of the region experienced decline in production. This was determined partly by geology and the location of exploitable deposits (Figure 2), but the distance to demand areas could have also influenced this trend. This observation is substantiated by results of spatial statistics calculations, that is, mean center and standard deviational ellipse statistics (Figure 6). The mean center of quarry locations weighted by their annual output (2010; black dot) was shifted east in comparison to the unweighted mean center of their locations (gray dot) by approximately 9900 m, whereas the weighted mean center location of quarries calculated for their 2018 output (red dot) was shifted further 2500 m eastwards.

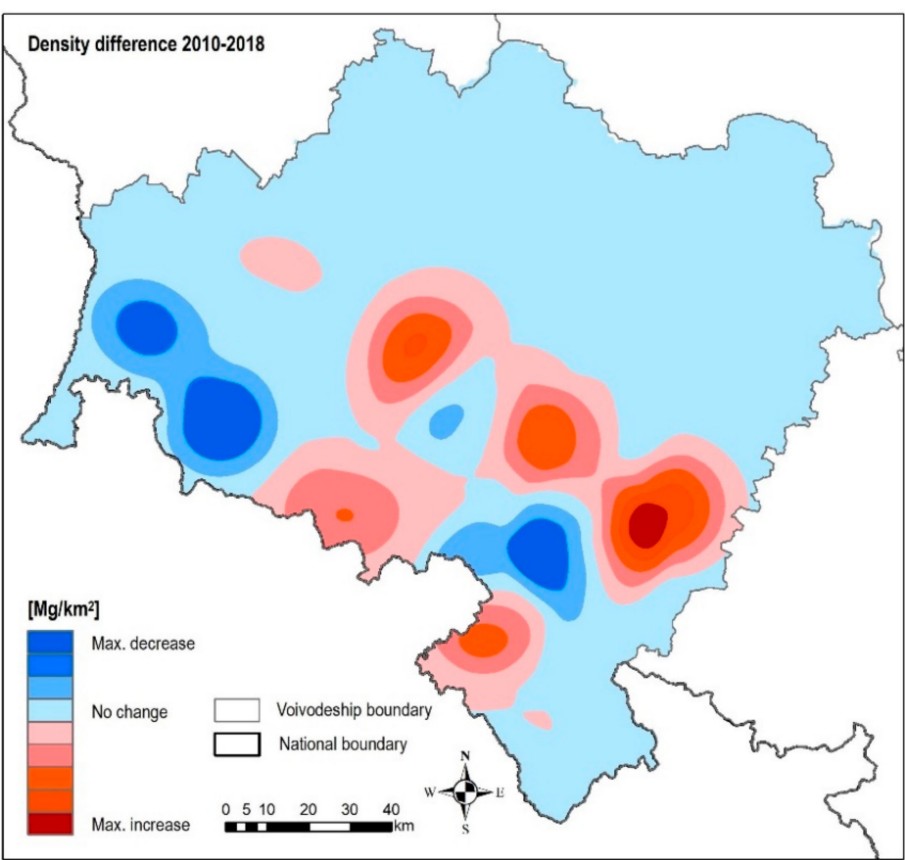

**Figure 5.** Change of the spatial density of dimension stones and crushed rocks production, 2010–2018.

The measure of the standard deviation of the features (quarries) from the mean center, major axis extending in the northwest–southeast direction and minor axis in the southwest–northeast direction, is the consequence of geological settings. The change in the shape of the weighted production ellipses (2010 and 2018) and different shape of the ellipses for the unweighted and weighted data also point to the slight eastward shift of production and concentration (smaller $x$ and $y$ axes of the 2018 ellipse compared to the 2010 ellipse) of rock raw material production.

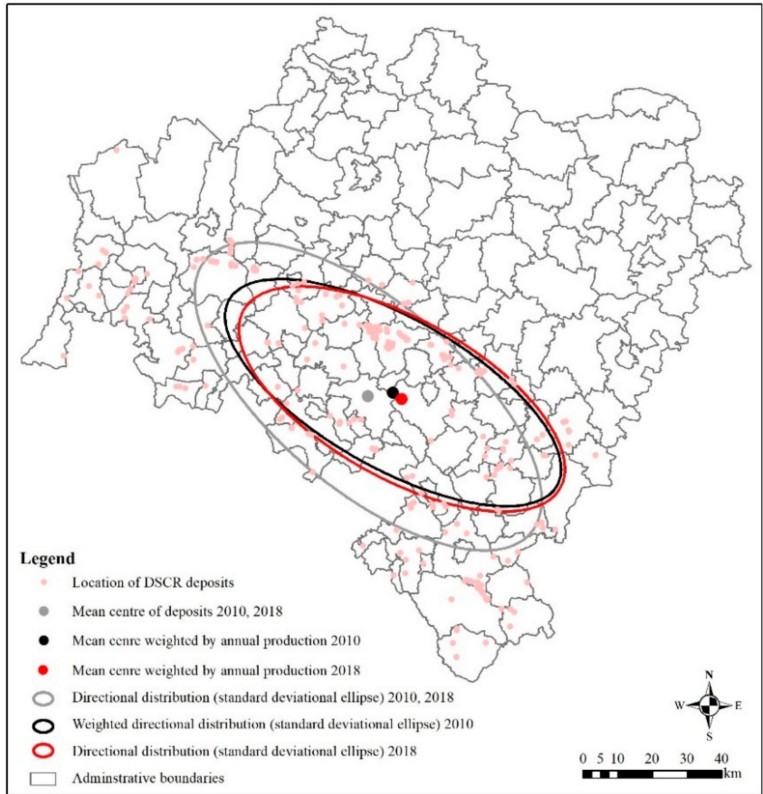

**Figure 6.** Mean center and standard deviational ellipse spatial statistics for dimension stones and crushed rock production in Lower Silesia, 2010–2018.

*4.3. Sufficiency of Dimension Stones and Crushed Rocks*

The results of this analysis indicate approximately how long, in years, the economic reserves of particular group of igneous and metamorphic rocks will last, as well as how long active quarrying operations can continue at the output level set for given scenarios. The reserve sufficiency of a given rock type (e.g., granite) was calculated, taking into account the total economic reserves that are the technically and economically extractable parts of economic resources according to a national classification [60]. Sufficiency of DSCR resources was analyzed in seven lithological groups: (1) basalt; (2) granite, syenite; (3) gabbro; (4) melaphyre, porphyry; (5) amphibolite (including migmatite, serpentinite and greenstones); (6) gneiss (including hornfels and metamorphic slate); and (7) marble. The results for the four selected scenarios are presented in Table 5 and are graphically presented in Appendix A Figure A1. Scenario A was calculated for the production levels of the last available data (2018), scenario B for maximum recorded output, scenario C for minimum recorded output, and scenario D for an average production (2010–2018).

**Table 5.** Sufficiency (in years) of igneous and metamorphic dimension stones and crushed rocks.

| Dimension Stones and Crushed Rocks | Economic Reserves [1] (Mg) | Sufficiency (Years) | | | |
|:---:|:---:|:---:|:---:|:---:|:---:|
| | | Scenario A | Scenario B | Scenario C | Scenario D |
| Basalt | 336,189 | 43 | 32 | 60 | 53 |
| Granite, syenite | 1,000,061 | 84 | 82 | 93 | 110 |
| Gabbro | 268,301 | 94 | 78 | 159 | 124 |
| Melaphyre, porphyry | 134,278 | 25 | 27 | 49 | 37 |
| Amphibolite [2] | 107,429 | 36 | 51 | 50 | 67 |
| Gneiss [3] | 340,348 | 91 | 76 | 189 | 139 |
| Marble | 226,005 | 389 | 389 | 386 | 380 |

1—sourced from the Polish Geological Institute [7]; 2—including migmatite, serpentinite, greenstones; 3—including hornfels, metamorphic slate.

### 4.4. Analysis of the Commune Potential for DSCR Mining Sector

Table 6 presents the results of AHP calculations to determine weights of the six criteria proposed and compared by the experts.

**Table 6.** Ranking of criteria based on weights from analytical hierarchy process (AHP) analysis.

| Rank | Criterion | Weight (%) |
|------|-----------|------------|
| 1. | vi. potential environmental conflicts | 30.0 |
| 2. | ii. undeveloped DSCR deposits of regional or national significance | 27.4 |
| 3. | iii. current DSCR quarrying operations | 19.9 |
| 4. | iv. railroad/DSCR loading point infrastructure | 8.8 |
| 5. | i. undeveloped DSCR deposits | 7.9 |
| 6. | v. revenue from mining taxes | 6.0 |

Among these criteria, the lack of environmental conflicts with location of documented DSCR deposits (meaning better access for development of a mining operation) was considered as the most important factor (weight of 30.0%), with slightly higher score than the occurrence of undeveloped DSCR deposits of regional or national significance (weight of 27.4%). Existing DSCR quarrying operations in the area were also considered to be a strong factor (weight of 19.9%). The remaining three criteria scored between 6.0% and 8.8%, and were considered of lesser significance by the experts. For six criteria ($n = 6$), the random index (RI) was equal to 1.24 [61], and the calculated resulting consistency ratio (CR) of the comparison matrix was 4.7%, which was below the suggested threshold value of 10%.

Each of the communes in the Lower Silesia region was assigned a value with respect to each of the six criteria considered. The values were allocated using a 1 to 6 point scale, where 6 meant the most favorable conditions and 1 the least favorable circumstances. When a given factor was not present in a commune, such as for undeveloped DSCR deposits or no deposits being available due to environmental constraints, a score of 0 was assigned and the commune was excluded from further analysis. There were 29 communes with undeveloped DSCR deposits of national or regional importance identified in the analysis. When these were considered further, we found that 22 of these communes had at least 1 documented DSCR deposit whose exploitation was not constrained by environmental protections requirements. The spatial congruence of deposit and nature protection areas (environmental conflict) was investigated with GIS map overlay analysis. Overall, nine communes meet all of the criteria and were investigated in full. First, weighted scores for the nine communes in each of the criteria were calculated, then weights of local preferences with respect to these criteria were determined, in accordance with the AHP methodology described earlier. The weights of local preferences with respect to criteria for these communes are presented in Table 7, and results of overall priority calculations (ranking) are given in Table 8 and in Figure 7. The consistency ratios (CR) for pairwise comparisons for alternatives with respect to criteria varied from 0.8% to 7.1% and were within the required limits (≤ 10%). The lowest value (0.8%) was obtained for the undeveloped DSCR deposit criterion, and the highest (7.1%) was for the railroad/DSCR loading point infrastructure criterion.

**Table 7.** Weights of local preferences with respect to criteria (i–vi).

| Alternatives (Communes) | i. | ii. | iii. | iv. | v. | vi. |
|---|---|---|---|---|---|---|
| Strzegom | 0.370 | 0.383 | 0.054 | 0.493 | 0.237 | 0.206 |
| Żarów | 0.105 | 0.255 | 0.194 | 0.161 | 0.118 | 0.317 |
| Strzelin | 0.105 | 0.061 | 0.030 | 0.083 | 0.118 | 0.042 |
| Niemcza | 0.105 | 0.075 | 0.030 | 0.044 | 0.118 | 0.025 |
| Męcinka | 0.079 | 0.045 | 0.194 | 0.044 | 0.118 | 0.025 |
| Dobromierz | 0.059 | 0.045 | 0.106 | 0.044 | 0.097 | 0.025 |
| Lwówek Śląski | 0.059 | 0.045 | 0.112 | 0.044 | 0.064 | 0.206 |
| Kłodzko | 0.059 | 0.045 | 0.026 | 0.044 | 0.064 | 0.129 |
| Bolesławiec | 0.059 | 0.045 | 0.256 | 0.044 | 0.064 | 0.025 |

**Table 8.** Results of overall priority calculations.

| Rank | Alternatives (Communes) | Overall Priority (%) |
|---|---|---|
| 1. | Strzegom | 26.4% |
| 2. | Żarów | 23.3% |
| 3. | Lwówek Śląski | 10.9% |
| 4. | Bolesławiec | 8.3% |
| 5. | Męcinka | 7.6% |
| 6. | Kłodzko | 6.9% |
| 7. | Strzelin | 5.8% |
| 8. | Dobromierz | 5.5% |
| 9. | Niemcza | 5.3% |

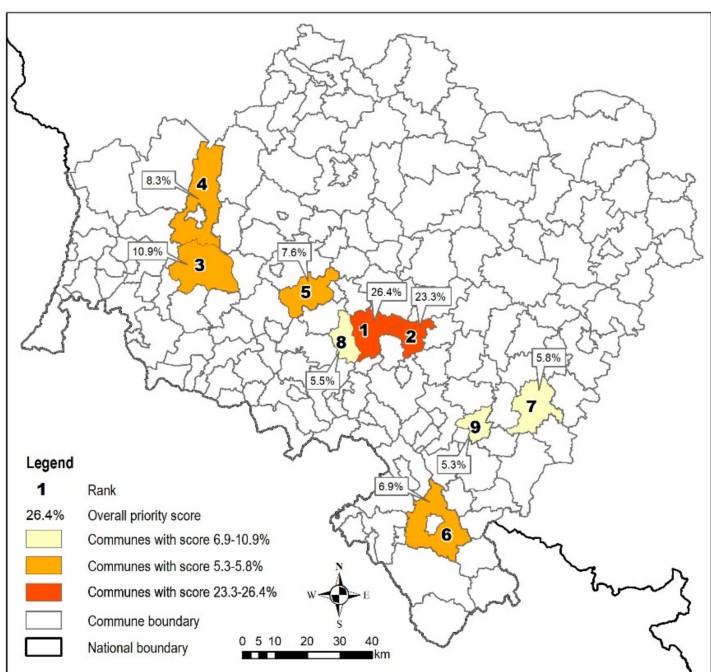

**Figure 7.** Map of communes with the greatest potential for development of DSCR mining sector.

## 5. Discussion

There is a limited number of published studies that attempt to measure spatial and temporal patterns of rock raw material quarrying in the regional (geographical) context and that focus on sustainable consumption and production patterns of these natural resources. The known research is predominantly based on analysis that employed standard descriptive statistics. In the case of Lower Silesia region, such a study has been done by Bem et al., who analyzed temporal changes of production

for the 2009–2014 period [62], as well as by Blachowski [12]. On the other hand, Danielsen and Kuznetsova performed a study that included statistical analysis of Norwegian aggregate industry in the European context, for the 2007-2013 period [18]. Balletto et al. worked on strategies to increase recycled aggregate use in Sardinia, Italy, after investigation of natural and recycled aggregates production with descriptive statistics. A more advanced approach was used by Bliss et al., who proposed a statistical- and Monte Carlo-based method to assess potential resources of natural aggregates in Idaho, USA [20].

The spatial context is usually taken into account in valorization of documented dimension stones and crushed rocks deposits or rock raw materials source areas, and GIS is utilized for map overlay of geological, social, economic, environmental, and other groups of criteria representing accessibility and potential of rock source area for development. Such an approach has been presented, for instance, in [13–17]. In some publications, multicriteria decision methodology is used for determining weights of particular criteria, such approach has been applied in investigating sites for waste disposal from rock raw material quarrying and sustainable management of mining waste, for instance, in [28,29]. GIS has also been used to process remote sensing and field survey data related to assessment of impacts associated with quarrying of rock raw materials, for instance, in [22–24,26]. Our study presents a different approach in its attempt to quantify and visualize the potential spatial pattern of dimension stone and crushed rock production and its change in time, as well as an analysis of the potential of particular administrative areas for sustainable development of the DSCR mining sector.

### 5.1. Analysis of Dimension Stone and Crushed Rock Production

The initial part of our research based on the descriptive statistics approach provided information on the general trend of dimension stone and crushed rock production. An attempt was made to correlate change in annual production of DSCR with annual GDP rates for Poland. For the 1998–2018 period, the calculated Pearson's correlation coefficient ($r = 0.645$) indicated significant positive correlation. Graph of annual GDP rate change and year to year production of DSCR change is presented in Figure 4. This is in accordance with conclusions from other studies that point to direct relationship between economic growth and demand for dimension stones and crushed rocks, as well as sand and gravel. This demand is highly variable and correlates with economic situation of particular countries [63,64]. Worldwide demand for all types of rock raw construction materials is estimated to be rising by 4.7% annually [18]. In the case of Lower Silesia, which is the main region of igneous and metamorphic rock raw material production in Poland with 89.5% share in case of igneous rocks and 99.0% share of metamorphic rocks (2018), the demand resulting from the developing economy and construction of new infrastructure is visible in data presented in Figures 3 and 4. Large quantities of rock raw materials are demanded by the booming road and building construction, as well as other sectors of economy. Lower Silesia is the main provider of granite, gabbro, and basalt; practically the sole producer of amphibolite, gneiss, and marble; and the dominant provider of melaphyre and porphyry. The highest peak in production (2011) was associated with construction of roads, stadiums, and other infrastructure for the European Football Championships of 2012. In 2012, a significant drop in production was recorded (−28.1%) from 38,370 thousand Mg to 27,689 thousand Mg. Steady level of production continued for the next 4 years, and in the 2 last analyzed years (2017, 2018) production increased year to year by +19.9% and 16.3%, respectively. It is worth noting that the Lower Silesia share of production of all types of DSCR raw materials, including sedimentary rocks, oscillates around 45% annually (44.1% in 2018). The remainder is produced in three other southern regions, namely, Swietokrzyskie with approximately 30% of production, Malopolskie with approximately 10%, and Slaskie with approximately 5% of production (sedimentary dimension stones and crushed rocks only). Among these regions, the Swietokrzyskie noted an increase of production in recent years [65]. Published statistical data indicate that 47.4% of total aggregate mining in Europe that accounts for 2.7 billion Mg per year is dimension stones and crushed rocks [64]. Poland is the seventh largest producer of DSCR among the 27 analyzed European countries, with a 5.6% share in 2017 and 0.4% point increase in the 2013–2017 period [5]. Among these

countries, 20 recorded increases of production occurred, most notably in Germany, Great Britain, Italy, Norway, Spain, and Sweden.

*5.2. Analysis of Spatial Distribution of Dimension Stone and Crushed Rock Production*

The spatial statistics calculated in GIS were aimed at identification of potential spatial patterns and their change in time in the Lower Silesia region. The density of DSCR production analysis, studied in [8] for an earlier period (2006–2010), was complemented here by mean center and standard deviational ellipses calculations. Results of both complementing approaches (Figures 5 and 6) indicated that the weight of production shifted towards the east. The map of the production difference between 2010 and 2018 (Figure 5) showed net increase of production per unit area in DCSR source areas located in the central and eastern parts of the region, and decrease in the peripheral western and southern parts, whereas spatial statistics maps (Figure 6) revealed movement of the mean center of quarry location weighted by their production. The GIS-based approach to analyze DSCR production allowed for the identification and quantification of a possible general spatial trend, as well as providing additional insight into the process of quarrying, for regional policy makers and administrators aiming at sustaining balanced development of these non-renewable resources. One of the reasons for this spatiotemporal change might be related to the attempt to reduce transport costs of rock raw materials to other parts of the country (demand areas). Transport costs from source to demand areas are related directly to the distance and one of the most significant final cost factors [15]. Another explanation might include the limiting effect of environmental and land use constraints for development of existing quarries and opening new ones in the southern parts of Lower Silesia that are characterized with a higher percentage of nature protection sites [8,17,19]. Furthermore, the noted change in spatial pattern might have a more complicated nature and be related to a greater extent with particular lithological types of rocks and their source areas.

*5.3. Sufficiency of Dimension Stones and Crushed Rocks*

According to previous studies carried out in Poland, on the country scale, the indicator of resource use for dimension stones and crushed rocks is equal to 32%, meaning that static sufficiency or these resources is approximately 40 years and practical, due to resource loss, might be 20% to 30% lower [56]. On the other hand, Lewicka and Burkowicz estimated economic reserves to last for approximately 50 years at the current levels of production [7].

The share of the domestic production in meeting the demand for these rock raw materials is over 99% [62]. Therefore, in this study several (four) scenarios were analyzed to estimate, in more detail, sufficiency of igneous and metamorphic DSCR in Lower Silesia for particular rock type groups. For basalt, the sufficiency ranged from 32 years for peak recorded yearly production to 60 years for lowest recorded output, with 53 years for an average production level. Sufficiency of granite reserves was estimated at 82 to 110 years, depending on the scenario; gabbro at 78 to 159 years; and gneiss at 76 to 189 years. The reserves of melaphyre and porphyry will last for just 25 to 49 years, and amphibolite for 36 to 67 years. Reserves of marble were the largest at the analyzed production levels (380 to 389 years). Considering the average production as the most realistic scenario that takes into account temporal variation in demand for rock raw materials, economic reserves of one group (melaphyre and porphyry) were found to be sufficient for less than 50 years, two groups (basalt, amphibolite) for about 50 years, another two groups (granite, gabbro) for around 100–150 years, and gneiss and marble rock raw materials for more than 150 years. These results indicate that analyzing sufficiency of rock raw materials reserves as a whole, as in the above cited studies, does not give the complete picture, and the availability of resources for the future varies with rock type and production levels (demand) taken into consideration. For example, comparing these values with the results of the previous study [12], new assessment indicates decrease in available melaphyre economic reserves and increase in amphibolite reserves. All of these estimates concern economic reserves in currently developed deposits and may change if new quarry operations are started. However, development of

new resources may be limited due to spatial constraints [17], and regional spatial development policy should be aware of the limited supply of these resources and be directed at enlarging the resource base of developed and valorization of undeveloped deposits because of the role of Lower Silesia in the supply of DSCR to the market. In addition, following the example of other regions, one should look towards increasing the share of recycled rock raw materials [18,27] in accordance with the concept of circular economy model [66]. As of yet, the reuse of DSCR, such as becoming a source of crushed rocks, is limited.

### 5.4. Potential for DSCR Mining Sector Development

This aspect of the study was aimed at identifying local communities (commune administrative level) that are the most recommended in order to sustain and develop the rock raw material mining sector of the economy. The analysis was based on multicriteria AHP methodology and six criteria listed in the results section. There were 29 communes investigated, and 9 of them were determined as having the highest potential. They were located in different parts of the region (Figure 7), but mostly in areas of high present-day production (Figure 5), such as communes ranked 1, 2, 7, 8, and 9 being associated mainly with granite and syenite source areas, and communes ranked 3 and 5 with basalt. Among the nine communes, Strzegom (1) and Żarów (2) have the highest potential for further development of DSCR quarrying. The ranking arises from high scores obtained in the all but criterion (iii) for the Strzegom commune and solid scores in all the criteria for the Żarów commune. The values of overall priority calculations for the remaining seven communes are lower, and among them the following four communes obtained the highest scores: Lwówek Śląski, Bolesławiec, Męcinka, and Kłodzko. These administrative units may also consider DSCR quarrying as a potential sector of the economy. It must be noted that the lowest values for the cases of Strzelin, Dobromierz, and Niemcza resulted from comparison with the other six communes and still indicate areas with high potential among the 169 communes in Lower Silesia. It must be noted that these results indicate preferences of a group of experts representing given specializations, and that a different group could result in other values of criteria weights and overall priority. However, the construction of AHP analysis attempts to objectify the preferences, and other scenarios should lead to generally the same results. The CR value indicates that there were no significant inconsistencies between the experts. It should be emphasized that more criteria could be considered in further and more detailed analysis, such as type, size, and quality of the documented DSCR resources. The proposed methodology is a different approach towards assessing accessibility of rock raw material resources than the weighted criteria map overlay presented in literature, such as that in [13–17]. These approaches could be used jointly, complementing each other to provide more reliable identification.

### 6. Conclusions

The research focused on analysis and assessment of igneous and metamorphic dimension stone and crushed rock quarrying in the Lower Silesia region, which is the chief supplier of these rock raw materials in Poland. The share of Lower Silesia production amounted to 89.5% for igneous rocks and 99.0% for metamorphic rocks. Attention was drawn to the spatial and temporal context of DSCR quarrying. The analyzed period covered 9 years (2010 to 2018). The proposed methodology relied on spatial statistics measures including density and directional distribution in geographic information systems to aid descriptive statistics usually used for such purposes. Multicriteria analytical hierarchy process approach was used for identification of communes prospective for development of quarrying. The proposed methodology could be used universally for other rock raw material source regions. The main findings of the research are (i) positive correlation of rock raw material production with annual GDP rates, and steady level of production following peak of 2011 with return to a growing trend in 2017; (ii) mapped shift of greatest intensity of quarrying in eastern direction, indicated by density and weighted mean center functions; (iii) estimated sufficiency of seven main lithological rock groups calculated for four scenarios of production levels that varies from less than 50 years for melaphyre

and porphyry (25–49), through approximately 50 years for basalt (32–60) and amphibolite (36–67), approximately 100 years for granite (82–110), and over 100 year for gabbro (78–159) and gneiss (76–189) to close to 400 years for marble (386–389); (iv) identification of local communities with greatest potential for development of the rock raw material industry on the basis of weighted analysis of identified factors such as occurrence of undeveloped DSCR deposits of regional or national significance, lack of potential conflicts with environmental protection requirements, tradition of quarrying, existing DSCR transport infrastructure, and share of mining taxes in local budgets. A multifaceted approach utilizing descriptive and spatial statistics in GIS, as well as AHP multicriteria analysis, provided additional insight into the scale and distribution of DSCR quarrying, including capture of the spatial context and information, upon which sustainable management of regional rock raw materials can be addressed in spatial policies and strategies.

**Author Contributions:** Conceptualization, J.B.; methodology, J.B.; software, J.B. and A.B.; validation, J.B.; formal analysis, J.B. and A.B.; investigation, J.B. and A.B.; resources, A.B.; data curation, A.B.; writing—original draft preparation, J.B. and A.B.; writing—review and editing, J.B.; visualization, J.B. and A.B.; supervision, J.B.; project administration, A.B.; funding acquisition, J.B. All authors have read and agreed to the published version of the manuscript.

**Funding:** The research has been partly supported by the statutory grant at the Department of Mining and Geodesy, Faculty of Geoengineering, Mining and Geology, Wroclaw University of Science and Technology.

**Acknowledgments:** The authors would like to acknowledge the Institute for Territorial Development, Lower Silesia Marshal Office for their help.

**Conflicts of Interest:** The authors declare no conflict of interest.

## Appendix A

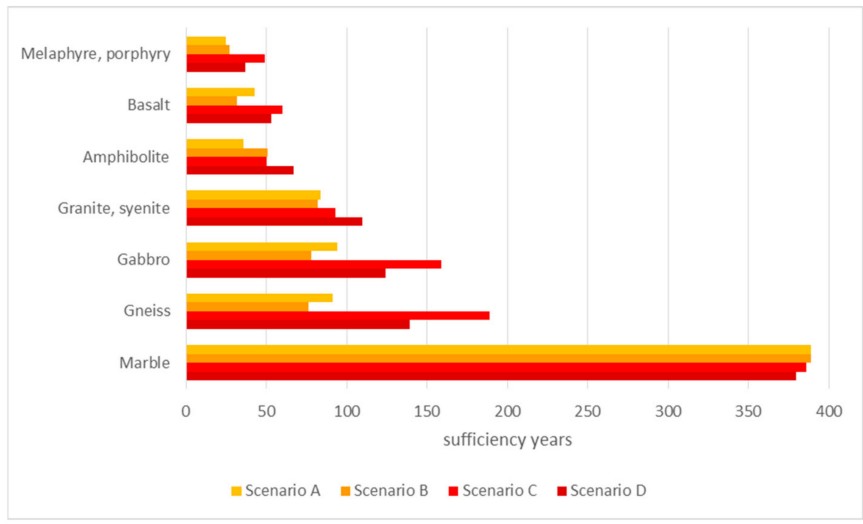

**Figure A1.** Graphical representation of reserves sufficiency for the analyzed DSCR groups for the four analyzed scenarios.

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
