# Peer review of "Spatial and Multicriteria Analysis of Dimension Stones and Crushed Rocks Quarrying in the Context of Sustainable Regional Development: Case Study of Lower Silesia (Poland)"

_sustainability, doi:10.3390/su12073022_

Round 1
Reviewer 1 Report
The research is focused aimed at analysis and assesment of spatial and temporal trends in distribution of reserves of production of igneous and metamorphic dimension stones and the crushed rocks in the Lower Silesia region which is the key source region of these rock raw materials in Poland.
Practical outcome of this study is a knowledge database for authorities, upon which sustainable management of regional rock raw materials can be based in the context of economic, social, and environmental impacts of their extraction.
Strenghts:
• The introduction of the paper provided sufficient background research and include all relevant references; the research has on long period of analyzing period covered 9 years (2010 to 2018).
• The proposed methodology is suitable and methods are adequately described; these relied on spatial statistics measures including density and directional distribution in geographic information systems to aid descriptive statistics usually used for such purposes.
• The identification of local communities with greatest potential for development of rock raw materials industry based on weighted analysis of important factors was based on mapping; mapped shift of greatest intensity of quarrying in eastern direction indicated by density and weighted mean center functions was obtained;
• The multifaceted approach utilizing descriptive and spatial statistics in GIS, as well as AHP multicriteria analysis provided additional insight into the scale and distribution of DSCR quarrying including capture of the spatial context and information, upon which sustainable management of regional rock raw materials can addressed in spatial policies and strategies.
Weeaknesses:
• The graphical aspects (Figures) do not have full English explanations and some of them are not clear enough; the Figure 1 is not enough clear, what about comment „1.564” on production explanation? Figure 7 has not enough explanation in Legend.
Reviewer 2 Report
The paper describe possibillity of regional development in once of the regions from Poland on the base production of igneous and metamorphic dimension stones and crushed rocks. Authors used AHP analysis to solve this problem. The text is compact and clear.
My comments is:
How many experts filled the questionnaire AHP method?
Explain the basis on which the criteria were adopted in table 6.
How the regional GDP (not global in Poland) depends from production of DSCR? Comparing with GDP (in Poland) the others product (f.e. builging production) will the similar. If GDP increase, the production of various goods is also growing, not only DSCR in Lower Silesia.
Table 8. Some describe is not complete in first row.
Reviewer 3 Report
Reason: The article is not fit for an international readership. It is a continuation of an article of 2014 of the senior author in Environmental Earth Sciences, 71: "Spatial analysis of the mining and transport of rock minerals (aggregates) in the context of regional development". This is a well written article and gives a good overlook about the Poland and Lower Silesia situation, the main source for Poland for aggregates.
The new submitted article for peer review (manuscript 757 699) would certainly be of interest to a Polish readership, let's say for a journal of the Polish Geological Society, for people who want to know as much detail as possible of aggregates in their own country. But why is it necessary to give more details for an international readership like the one of "Sustainability". Is it of interest for a reader for example in Canada or Australia, after being already informed in the 2014 article about the resource sufficiency of aggregate deposits in Lower Silesia, how the detailed resource sufficiency is for the various types of aggregates like basalt, granite gabbro etc.? It would only be of interest, if there are special uses like for granite aggregates or basalt aggregates. But normally all aggregates can substitute each other. And the authors do not give any special applications or other reasons which would justify the split for details for an international readership.
(In fairness, it shall be mentioned that the term "resource sufficiency" is an improvement to the term "reserve sufficiency, used in the 2014 article. One wonders however, how it is possible that in the 2014 article in Fig. 7 the majority of the deposits (about 43) has reserve sufficiency of more than 100 years, but in Table 5 of the manuscript in the Scenario A and B only marble has a resource sufficiency of more than 100 years and in Table 3 for the years 2010 to 2015 the number of marble deposits is given with only 22. In addition, in the 2014-paper the documented dimension and crushed stone deposits is given with 272, but in the submitted manuscript the number of deposits in the years 2014 and before is only 222 to 224.)
To conclude, the paper should be rewritten and concentrate on aspects which are new and not dealt with in the 2014-paper and which are of interest for an international readership. This is, for example, the shifting mean center and standard deviational ellipse spatial statistics (Fig. 6). But what does it mean? This should be elaborated in far more detail than just say in line 445 that it might be related to transport costs. Far more detail about tansport is available already in the 2014 paper. In Germany, for example, the order of aggregates cost can be about 20 @/t, transport cost by truck 0.20 €/km. So a 10 km shift could mean on average a decrease or increase of about 10% of the costs.
Detailed comments:
1.) The authors use a peculiar way of referencing, e.g. "[x] states that... ". This should be changed to: (Author name) states that...........[x]. with quote at the end.
2.) In line 37 sedimentary rocks are mentioned, but not again until line 422, where we learn that there are also sedimentary aggregate deposits in Lower Silesia. You have to explain in the beginning, why you omit them.
3.) The map in Fig. 1 has to be redrawn for an international readership. Give the neighbours like in the key map of Fig. 2., prolongate the Baltic coast line into Germany and Russia, so reader can realize that this is coastline in contrast to a boundary and introduce the name "Baltic Sea".
In English, all proper name like for woiwodeships are given with a capital letter.
For an international readership it would be interest to see where the captial Warsaw is located.
4.) line 91 and 108: "mosaic" according the the Oxford Dictionary is a noun.
5.) Fig. 2: You have to explain in the caption what Nature 2000 SPAs and SACs are.
6.) line 109: These are not elements but rock types
7.)Line 112: Not "inconsistently", but "with an unconformity""
8.) line 193: S (Index t) is wrong. It has the dimension [years] and is the resource sufficiency in your terminology.
Reviewer 4 Report
- When you talk about ornamental/dimension stones and crushed rocks, you should talk about “quarrying” instead of “mining”. “Mining” is referred to ore minerals.
- Introduction: I suggest inserting more bibliographical references relating to the types of ornamental stones and crushed stone at world and European level. It would also be desirable to separate the ornamental/dimension stones from crushed rocks. Please indicate at least the most important commercial names of Polish dimension stones quarried in this region. Why don’t you consider also sand and gravel from alluvial deposits? Of course, it’s not crushed rock, but in many cases uses are similar.
- Figure 1: please insert scale bar, and eventually the position of Poland within Europe.
- Geology and mineral resources of Lower Silesia: there are few bibliographic references for the geological set-up of the region. Lines 106 – 108: please specify age, not just “older” and “younger”. It would be nice to separate “dimension stones” from “crushed rocks” (much lower value). Can you indicate geological parameters (fracture density, lithological and textural homogeneity) to distinguish between dimension stones (produced in blocks) and crushed stone? Are there dimension stone quarries that “recycle” quarry waste in crushed rock?
- The term “melaphyre” is obsolete…what do you mean? Altered basalts? Basic volcanic rocks?
- “crystallic” slate is inappropriate, what do you mean?
- Figure 5: insert scale bar and orientation;
- Figure 7: insert scale bar and orientation;
- Discussion and conclusion: although the mathematical and statistical treatment is adequate, there are no references to the geological and structural characteristics of the deposits, which are decisive for defining the quality of the material and the use (dimension stone or crushed rock). In a circular economy perspective, it would also be useful to consider the possible recovery of dimension stone quarry and laboratory waste as crushed rock or for other purposes (e.g. recovery of quartz and feldspars).
- Mg could also be indicated as metric tons, for simplicity.
Round 2
Reviewer 3 Report
Concerning my point 1: Comments on the resource sufficiency, as well as the difference in numbers of deposits.
Re: Explanation why the number of deposits in Table III differ from the number in Blachowski, Environ. Earth Science 2014. Give at Table III (heading or below) an explanation for the reader as you do in the "Response to Reviewer 3 Comments". Concerning your term "deposits with no economic significance": This a wrong nomenclature, a contradiction in itself. "Deposits" are mineral accumulations which can be economically exploited or at least are potentially exploitable. You have to say "occurrences with no economic significance". To counteract comments of readers who could ask, why you included them at all in your 2014 paper, you could say "with today's knowledge occurrences with no economic significance".
Reviewer 4 Report
The authors made all the suggested improvements.Author Response
Reply to review #4.
Thank you for the positive comment.
Sincerely yours
Jan Blachowski